# Carbon-Ion Radiotherapy for Prostate Cancer in Patients with a History of Surgery for Benign Prostatic Hyperplasia

**DOI:** 10.3390/cancers17183039

**Published:** 2025-09-17

**Authors:** Atsushi Okato, Kosei Miura, Tomoki Yamaguchi, Mio Nakajima, Hirokazu Makishima, Takanobu Utsumi, Koichiro Akakura, Hiroyoshi Suzuki, Masaru Wakatsuki, Hiroshi Tsuji, Tomohiko Ichikawa, Hitoshi Ishikawa

**Affiliations:** 1QST Hospital, National Institutes for Quantum Science and Technology, Chiba 263-8555, Japan; okato.atsushi@qst.go.jp (A.O.); nakajima.mio@qst.go.jp (M.N.); makishima.hirokazu@qst.go.jp (H.M.);; 2Department of Urology, Toho University Sakura Medical Center, Chiba 285-8741, Japan; takanobu.utsumi@med.toho-u.ac.jp (T.U.); hiroyoshi.suzuki@med.toho-u.ac.jp (H.S.); 3Department of Urology, Japan Community Healthcare Organization Mishima General Hospital, Shizuoka 411-0801, Japan; akakurak@ae.auone-net.jp; 4Particle Radiotherapy Clinic, Tokyo 102-0094, Japan; tsuji.hiroshi@qst.go.jp; 5Department of Urology, Japan Community Healthcare Organization Funabashi Central Hospital, Chiba 273-8556, Japan; tomohiko_ichikawa@faculty.chiba-u.jp

**Keywords:** carbon-ion radiotherapy, prostate cancer, benign prostatic hyperplasia

## Abstract

The safety of carbon-ion radiotherapy for prostate cancer in patients with a history of surgery for benign prostatic hyperplasia remains unclear. This study evaluated the long-term outcomes in patients who underwent carbon-ion radiotherapy. The treatment resulted in favorable cancer control and an acceptable incidence of urinary complications. However, an increased risk of hematuria was observed in patients who underwent transurethral resection of the prostate or radiotherapy shortly after surgery. This study validates carbon-ion radiotherapy as a viable treatment method in this specific population and highlights the importance of individualized treatment planning and long-term monitoring, ultimately improving management of radioresistant malignancies such as prostate cancer.

## 1. Introduction

Prostate cancer is a primary global health concern, ranking as the fourth most frequently diagnosed cancer overall and the second most common among men worldwide, with approximately 1.47 million new cases recorded in 2022 (7.3% of all cancers) [1]. Radiotherapy (RT) is the standard treatment for localized disease. Technological advancements have improved dose conformity and reduced adverse effects [2]. Among these, carbon-ion radiotherapy (CIRT) has emerged as a highly advanced technique that offers both precise dose distribution and enhanced biological effectiveness. Its unique physical and biological characteristics, including the Bragg peak and high relative biological effectiveness (RBE), enable high-dose delivery to tumors while minimizing exposure to adjacent healthy tissues, ultimately improving the management of radioresistant malignancies such as prostate cancer. These advantages have led to the increasing adoption of CIRT, particularly in high-risk or anatomically complex cases [3,4,5].

Despite favorable outcomes reported in the general population, the safety profile of CIRT in specific clinical settings remains elusive, particularly in patients with a history of surgery for benign prostatic hyperplasia (BPH). Notably, surgical procedures for BPH, including transurethral resection of the prostate (TURP) and holmium laser enucleation of the prostate (HoLEP), are commonly performed in older men with lower urinary tract symptoms [6,7,8]. However, these interventions result in substantial anatomical changes—such as resection cavities, tissue fibrosis, and altered vascular structures—that may render the genitourinary (GU) tract more vulnerable to radiation-associated complications, including hematuria, incontinence, and urethral stricture. In this surgically altered population, the bladder neck and periurethral tissues are particularly fragile, leading to a higher risk of treatment-related complications. Therefore, beyond tumor control, supportive strategies—including dose optimization around the bladder neck/urethra, appropriate use of fiducial markers and hydrogel spacers when feasible to reduce rectal exposure, and lifestyle guidance to avoid constipation and bladder irritants—are essential components of CIRT planning and delivery in this population. Furthermore, postoperative adhesions around the prostate may complicate salvage surgical procedures, and concerns over radiation-related adverse outcomes have led some institutions to avoid offering curative treatments.

TURP-related adverse outcomes have been reported not only in RT but also in surgical interventions for prostate cancer [9,10]. Several studies have demonstrated that patients who undergo radical prostatectomy after TURP—either via open or laparoscopic approaches—are likely to experience increased risks of intraoperative bleeding, bladder neck strictures, and positive surgical margins [11,12,13]. These findings suggest that anatomical alterations after BPH surgery lead to complications across multiple treatment modalities, emphasizing the need for careful assessment when selecting curative strategies.

In conventional RT settings (such as intensity-modulated radiation therapy and stereotactic body radiotherapy), a history of BPH surgery has also been consistently associated with a higher incidence of late GU complications [14,15]. Contributing factors include the time interval between surgery and RT, tissue resection extent, and radiation dose to surrounding healthy tissues [16,17]. Consequently, clinicians often approach RT with caution in this patient subgroup. Moreover, limited evidence exists regarding the safety of CIRT in this context, and it remains unclear whether its dosimetric precision and favorable biological properties can mitigate the risks associated with altered anatomy. Given the increased adoption of CIRT and the aging prostate cancer population, determining the safety profile of CIRT for patients with a history of BPH surgery is crucial. Thus, addressing these issues is essential not only for appropriate patient selection but also for informing treatment planning and patient counseling.

This single-center retrospective study aimed to evaluate the long-term safety of CIRT in patients with prostate cancer and a history of BPH surgery. Specifically, we assessed the incidence and severity of late GU complications and examined oncological outcomes, including biochemical recurrence-free survival (bRFS) and local control. Although we evaluated treatment efficacy, the primary focus of this analysis was on safety and tolerability. By addressing a critical gap in the literature, this study sought to support the safe and effective use of CIRT in patients with surgical modifications, thereby expanding access to high-precision, curative-intent RT.

## 2. Materials and Methods

### 2.1. Patients and Study Design

This retrospective cohort study evaluated the safety and efficacy of CIRT for prostate cancer in patients with a history of transurethral or open adenomectomy for BPH. The study was approved by the Institutional Review Board of QST Hospital (approval number: N24-022) and was publicly disclosed on the institution’s website. All potential participants were given the option to opt out. An opt-out approach was employed instead of written informed consent, and all data used in this study, including cystoscopy images, were fully anonymized with no identifiable information. Many patients were referred to our institution after being deemed unsuitable for radical prostatectomy or photon-based radiotherapy at other hospitals. The reasons for selecting CIRT included advanced age, comorbidities, unfavorable anatomical conditions following BPH surgery, or patient preference for a non-surgical modality. Eligible patients included those diagnosed with prostate cancer who underwent CIRT in combination with androgen deprivation therapy (ADT). Surgical treatment for BPH before CIRT was performed in patients who were unresponsive to medical therapy or in those presenting with severe lower urinary tract symptoms. Surgical approaches included TURP, which involves the resection of obstructive prostatic tissue using an electrocautery loop; transurethral enucleation using bipolar energy (TUEB); and laser-based procedures, such as HoLEP and photoselective vaporization of the prostate (PVP), which are considered less invasive alternatives. In selected cases, simple suprapubic prostatectomy (SPP) was performed via an extraperitoneal approach to enucleate the enlarged adenomas. In addition, transurethral microwave therapy (TUMT), a minimally invasive procedure that uses microwave energy to induce thermal necrosis of the prostatic tissue, was also employed. In patients with a history of multiple BPH surgeries, the most recent procedure was considered for categorization of the surgical technique.

### 2.2. CIRT

CIRT was administered according to previously established protocols. Carbon-ion beams were delivered once daily, 4 days per week, targeting the prostate and seminal vesicles. Radiation doses were expressed in Gy (RBE-weighted dose based on the modified microdosimetric kinetic model) [18]. Dose-fractionation regimens included 57.6 Gy in 16 fractions, and 51.6 Gy or 54.0 Gy in 12 fractions. Target volumes were delineated on simulated computed tomography (CT) or fused CT-magnetic resonance imaging images.

### 2.3. ADT

ADT was administered based on the National Comprehensive Cancer Network (NCCN) risk classification. Treatment comprised subcutaneous injections of luteinizing hormone-releasing hormone (LH-RH) analogs, with or without oral antiandrogens. Patients with low-risk diseases did not undergo ADT. Intermediate-risk patients underwent neoadjuvant ADT for 3–7 months in combination with CIRT, whereas high-risk patients received ADT for 24 months, including both the neoadjuvant and adjuvant phases.

### 2.4. Follow-Up and Endpoint Assessment

Patients were followed up every 3–6 months within the first 5 years post-CIRT and every 6–12 months thereafter. Biochemical recurrence was defined according to the Phoenix definition (nadir prostate-specific antigen [PSA] + 2.0 ng/mL), excluding transient PSA elevations attributable to benign causes or fluctuations within the threshold. bRFS was calculated from the time of CIRT or neoadjuvant ADT initiation. Adverse events (AEs) were assessed through clinical interviews, laboratory tests, and endoscopic evaluations and categorized as acute or late relative to CIRT initiation. Acute events were defined as those occurring within 3 months, and late events as those occurring thereafter. Events were graded according to the Common Terminology Criteria for Adverse Events (CTCAE) v5.0.

### 2.5. Statistical Analysis

Survival outcome was estimated using the Kaplan–Meier method. Group comparisons were performed using univariate and multivariate logistic regression analyses. Statistical significance was set at *p* < 0.05. All statistical analyses were performed using the IBM SPSS Statistics v27 (IBM, Armonk, NY, USA) and EZ-R (v1.81; based on R v4.5.0, The R Foundation for Statistical Computing, Vienna, Austria) software (Jichi Medical University, Tochigi, Japan) [19].

## 3. Results

### 3.1. Patient Characteristics

Between 2007 and 2023, 3848 patients with prostate cancer underwent CIRT at our institution, of whom 74 (1.9%) had previously undergone surgery for BPH. The median patient age was 67 years at the time of BPH surgery and 75.5 years at CIRT initiation, with a median interval of 8 years between the two interventions. According to NCCN risk groups, 4.1% were low-risk, 48.6% intermediate-risk, and 47.3% high-risk. Prior BPH procedures included TURP in 66.2% of patients and HoLEP in 20.3%, with SPP, TUEB, PVP, and TUMT performed less frequently; three patients underwent multiple procedures. The baseline characteristics of the 74 patients are summarized in Table 1.

### 3.2. Adverse Events (AEs)

#### 3.2.1. Acute AEs

CIRT was well tolerated acutely. Grade 2 GU toxicity was observed in 4 of 74 patients (5.4%): urinary frequency in three patients (4.1%) and dysuria in one (1.4%). No Grade ≥ 3 acute GU events were observed. Acute GI toxicity was negligible, with no Grade ≥ 2 events. The overall incidence rates of grade 2–3 acute and late GU and GI events are summarized in Table 2.

A detailed, event-specific breakdown of all acute Grade 2–3 GU and GI toxicities is presented—alongside the corresponding late events—in Table 3.

#### 3.2.2. Late AEs

Late Grade ≥ 2 GU toxicity was observed in seven patients (9.5%): Grade 2 hematuria in three patients (4.1%), urinary frequency in one (1.4%), urethral stricture in one (1.4%), and Grade 3 hematuria in one (1.4%). Several cases of gross hematuria developed months to years after CIRT, and urethral strictures required intervention. Late GI toxicity was infrequent, with only one case (1.4%) of Grade 2 rectal hemorrhage, and no Grade ≥ 3 GI events. The late event data are also included in Table 3. In addition, among the less common procedures, one patient with a history of SPP experienced Grade 1 urinary frequency, one patient after PVP developed Grade 1 hematuria, and one patient after TUEB also presented with Grade 1 hematuria.

#### 3.2.3. Cystoscopic Findings in Patients with Hematuria

On cystoscopy, gross hematuria was invariably associated with mucosal telangiectasia, consistent with radiation cystitis (Figure 1A–C). All illustrative cases involved patients who had previously undergone TURP.

### 3.3. Hematuria Risk and Time-to-Event Analyses

#### 3.3.1. Risk-Factor Analysis for Hematuria

To identify the clinical predictors of hematuria following CIRT in patients with a history of BPH surgery, we performed univariate and multivariate logistic regression analyses for both Grades ≥ 1 and ≥2 hematuria. The results are summarized in Table 4 below.

For Grade ≥ 1 hematuria, univariate analysis revealed no significant predictors. However, a shorter interval between BPH surgery and CIRT showed a non-significant trend toward increased bleeding risk (odds ratio [OR] = 0.93 per year, 95% confidence interval [CI]: 0.85–1.02, *p* = 0.126), and TURP compared to HoLEP showed a similar trend (OR = 2.60, 95% CI: 0.61–18.0, *p* = 0.246). In multivariate analysis, both factors emerged as independent predictors: interval from surgery (OR = 0.87, 95% CI: 0.75–0.98, *p* = 0.041) and TURP (OR = 10.3, 95% CI: 1.41–127, *p* = 0.037). Other variables, including age at CIRT, total radiation dose, diabetes mellitus, anticoagulant therapy, and NCCN risk group, were not significant.

To further evaluate the predictive utility of the interval between BPH surgery and CIRT, a receiver operating characteristic (ROC) analysis was conducted. The area under the curve (AUC) was 0.63, indicating modest discriminative ability. The optimal cutoff value, determined using the Youden index, was 6.5 years, which yielded a 66.7% sensitivity and a 66.1% specificity. These findings suggest that patients receiving CIRT within 6.5 years post-BPH surgery may be at a higher risk of developing hematuria. The ROC curve is shown in Figure 2.

For Grade ≥ 2 hematuria, seven events were observed. In univariate analysis, diabetes mellitus was the only significant predictor (OR = 9.0, 95% CI: 1.09–74.2, *p* = 0.041), whereas anticoagulant therapy showed a non-significant trend toward increased risk (OR = 5.36, 95% CI: 0.68–42.2, *p* = 0.111). Although diabetes mellitus retained the strongest signal (OR = 9.64, 95% CI: 1.46–14.7, *p* = 0.068), multivariate analysis identified no significant predictors. Other variables, including TURP, age at CIRT, total dose, interval from surgery, and NCCN risk group, were not associated with Grade ≥ 2 hematuria.

#### 3.3.2. Time-to-Event Analysis of Hematuria

Kaplan–Meier analysis demonstrated that the cumulative probability of freedom from hematuria gradually decreased over time, reaching approximately 75% at 5 years (Figure 3).

Subgroup analyses showed a non-significant trend toward a higher incidence of hematuria in patients with a history of TURP compared with HoLEP (5-year hematuria-free probability: ~70% vs. ~90%, log-rank *p* = 0.18; Appendix A). Similarly, patients receiving anticoagulant therapy tended to have slightly lower hematuria-free probabilities than those not receiving anticoagulants, although the difference was not statistically significant (5-year probability: 70% vs. 78%, log-rank *p* = 0.38; Appendix A).

### 3.4. Cumulative Incidence of Grade ≥ 2 GU AEs

To provide a more comprehensive perspective on the overall safety profile of CIRT in this patient population, we evaluated the cumulative incidence of all Grade ≥ 2 GU AEs—including, but not limited to, hematuria, dysuria, urinary frequency, and urinary retention—using the Kaplan–Meier method. At 12, 24, and 36 months post-CIRT, the cumulative incidence rates for these GU events were 0.0%, 1.4%, and 5.0%, respectively. The curve gradually increased without a distinct inflection point, and the overall rate consistently remained <10% throughout the follow-up period, indicating that severe GU toxicity post-CIRT in patients with a history of BPH surgery was relatively uncommon (Figure 4).

### 3.5. Oncological Outcomes

Oncological outcomes were favorable across all risk groups. The 5-year bRFS rate was 100% in both the low- and intermediate-risk groups, with no recurrences observed. In the high-risk group, the 5-year bRFS was 88.6% [0.749, 0.991], indicating that most patients remained biochemically disease-free 5 years post-treatment, with only a few experiencing recurrences. These results suggest excellent midterm tumor control following CIRT in patients with a history of BPH surgery, particularly in low- and intermediate-risk groups. Outcomes in high-risk patients were also favorable. The Kaplan–Meier curves illustrating bRFS by risk group are shown in Figure 5.

## 4. Discussion

This study evaluated the safety and effectiveness of CIRT in patients with prostate cancer who had undergone surgical intervention for BPH, including TURP and HoLEP. With an aging population, prostate cancer is becoming increasingly prevalent in individuals with prior BPH surgery. Despite their clinical relevance, such patients have been underrepresented in radiation oncology studies, primarily due to concerns regarding treatment-associated GU complications. Studies have reported that transurethral procedures are associated with various urinary sequelae, including urethral stricture (2.2–9.8%), bladder neck sclerosis (0.3–9.2%), and urinary incontinence (1–6%) [20,21,22,23]. In addition, the risk of developing prostate cancer following BPH surgery has also been reported [24,25]. These findings underscore the need for a careful assessment of radiation-related safety in this unique patient cohort. Anatomical alterations following BPH surgery, including changes in mucosal surfaces, vascular structures, and bladder neck morphology, likely predispose tissues to subsequent radiation-induced injuries. Surgical sequelae such as fibrosis and adhesions may also complicate RT planning and delivery [26,27]. Notably, many patients in our cohort were referred after being deemed unsuitable for photon-based RT or salvage surgery.

Despite these potential vulnerabilities, our findings indicate that CIRT is generally well-tolerated in this population. Our analysis demonstrated a low cumulative incidence of Grade ≥ 2 hematuria, with rates of 1.4% and 5.0% at 24 and 36 months, respectively, and no cases within the first 12 months. This gradual increase suggests a delayed onset of clinically significant hematuria, rather than immediate post-treatment effects. Overall, Grade ≥ 2 late GU events were observed in 8.1% of patients, including one case (1.4%) of Grade 3 hematuria. Although these rates (8.1% in our cohort) are modestly higher than those reported in large-scale CIRT registries, such as the J-CROS study (5.0%) [28], they remain acceptable considering the anatomical complexity of the cohort.

Cystoscopy consistently revealed mucosal erosion and dilated capillaries at the bladder neck of patients presenting with hematuria, suggesting that previously operated fields are particularly susceptible to radiation-related vascular injury and impaired tissue regeneration. Furthermore, differences in hematuria incidence were observed between the surgical modalities, with a higher rate among patients who underwent TURP than among those who underwent HoLEP. Histopathological studies have shown that HoLEP confines thermal injury to a narrow rim of approximately 0.4–1 mm, thereby minimizing collateral damage to deeper prostatic tissues and vasculature [23]. The holmium:YAG laser used in HoLEP is highly absorbed by water, allowing precise tissue ablation and immediate coagulation of small vessels without carbonizing the vessel wall. These properties enhance vascular elasticity preservation and effective hemostasis [29]. In contrast, TURP uses monopolar or bipolar electrocautery, which induces deeper coagulative necrosis (approximately ≥ 1.5 mm) and may lead to broader tissue damage, postoperative scarring, and vascular recanalization [23,30]. These differences in tissue response may explain the higher incidence of delayed bleeding and radiation-induced hematuria observed in patients with a history of TURP. In addition to surgical modality, patient factors also warrant consideration. Our cohort had a higher median age than that reported in previous CIRT studies [28,31,32]. Age-related physiological changes, such as impaired vascular integrity, reduced tissue regenerative capacity, and increased susceptibility to radiation injury, might increase the risk of adverse outcomes [33,34]. Although no significant predictors for Grade ≥ 2 hematuria were identified, likely due to the limited number of events, exploratory analyses for Grade ≥ 1 hematuria suggested possible trends. Diabetes mellitus also showed a non-significant tendency toward more severe hematuria, a plausible association given the known microvascular pathology and impaired healing associated with diabetes [35]. Collectively, these findings emphasize the importance of surgical history, age, and comorbidities in risk stratification and individualized management.

Given these risks, supportive strategies should be incorporated into treatment planning to minimize urinary morbidity and preserve long-term function. Dose optimization around the bladder neck and periurethral region is essential, particularly in patients with prior TURP. Practical measures, such as the appropriate use of fiducial markers and hydrogel spacers (primarily to reduce rectal exposure and facilitate conformal planning in anatomically altered pelves) [36,37], along with lifestyle guidance—including the avoidance of constipation and bladder irritants—may further reduce the risk of hematuria and contribute to maintaining quality of life. Encouragingly, gastrointestinal toxicities were rare, with only one case of Grade 2 proctitis and no severe events, underscoring the favorable dosimetric properties of CIRT in this anatomically complex setting.

While our findings provide objective evidence of safety, patient-reported outcomes (PROs) such as urinary function, quality of life, and decision regret were not evaluated in this study. These measures are increasingly recognized as essential for capturing the full impact of treatment beyond physician-reported toxicities [38]. In addition, objective parameters such as post-void residual urine volume (PVR) were not systematically assessed. Previous external beam radiation therapy (EBRT) studies have suggested potential declines in bladder capacity and compliance without major PVR changes [39], and a prospective CIRT study has also reported longitudinal changes in urinary function and prostate volume [40], underscoring the value of incorporating such assessments in future analyses. It should also be noted that Grade 1 hematuria in this study was primarily based on patient self-reports of gross hematuria. However, because this was a single-institution study with standardized follow-up protocols, variability in the assessment of such events was likely minimized. Moreover, the delayed onset of hematuria observed in our cohort underscores the importance of long-term follow-up. Regular symptom assessment and timely cystoscopy, when indicated, are necessary to detect and manage late-onset morbidity.

Oncological outcomes appeared favorable. The 5-year bRFS rates were 100% in the low- and intermediate-risk groups and 88.6% in the high-risk group, comparable to or slightly exceeding outcomes of patients without prior BPH surgery [28]. These results are broadly consistent with large Japanese series such as J-CROS and with randomized trials in Western populations (ProtecT and SPCG-4), which have shown comparable long-term outcomes across surgery, radiotherapy, and active monitoring [41,42]. Furthermore, the Surveillance, Epidemiology, and End Results Program (SEER) registry data confirm that prostate cancer is the most diagnosed cancer among men in the United States, providing additional context for the international relevance of our findings [43]. The NCCN guidelines further emphasize radiotherapy as a standard curative option across risk groups, supporting the applicability of our results in an international context [44]. Recent systematic reviews further contextualize our findings. Brachytherapy boost combined with EBRT has shown favorable oncological efficacy but is associated with increased GU/GI morbidity [45]. Similarly, stereotactic body radiotherapy (SBRT) re-irradiation for radio-recurrent prostate cancer has demonstrated promising tumor control with acceptable safety, underscoring the potential of highly conformal modalities in complex clinical scenarios [46].

This study has several limitations, including its retrospective single-institution design, small cohort size, and relatively short follow-up. Moreover, the absence of a comparison group without prior BPH surgery limits generalizability, and parameters such as PVR and PROs were unavailable. These factors collectively warrant cautious interpretation.

## 5. Conclusions

CIRT provides excellent biochemical control and acceptable late toxicity in patients with prostate cancer and a history of BPH surgery. The cumulative incidence of Grade ≥ 2 GU-AEs remained low (5% at 36 months), and severe events were rare, underscoring the feasibility of CIRT even in anatomically altered pelvic fields. Individualized treatment planning that accounts for age-related frailty and prior surgical anatomy, alongside vigilant long-term surveillance for delayed urinary complications, is essential. Overall, these results support CIRT as a promising curative option in this surgically altered population. However, the absence of a non-BPH control group necessitates cautious generalization, and confirmation in larger prospective cohorts incorporating functional outcomes will be necessary to establish its role more definitively.

## Figures and Tables

**Figure 1 cancers-17-03039-f001:**
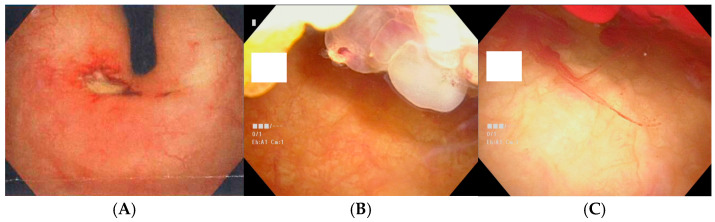
Cystoscopic findings of radiation-induced bladder mucosal changes in patients with gross hematuria: (**A**) Inverted view of the bladder neck showing bleeding source from telangiectatic mucosa. (**B**) Polypoid regrowth of prostatic tissue at the bladder neck following TURP. (**C**) Active bleeding from the polypoid lesion, consistent with radiation cystitis. All cases had a history of TURP. TURP, transurethral resection of the prostate.

**Figure 2 cancers-17-03039-f002:**
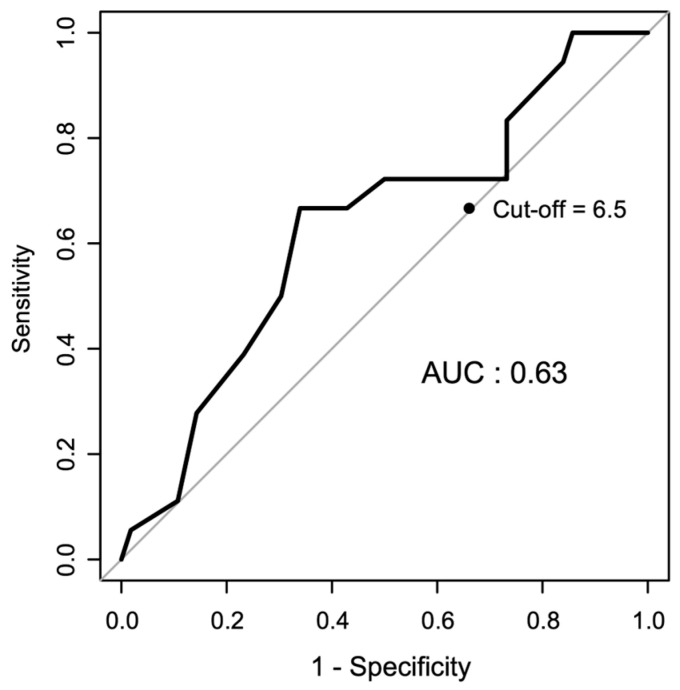
ROC analysis for predicting hematuria based on the interval between BPH surgery and CIRT. AUC was 0.63, indicating modest discriminative ability. The optimal cutoff value, identified using the Youden index, was 6.5 years, with a 66.7% sensitivity and a 66.1% specificity. AUC, area under the curve; ROC, receiver operating characteristic; BPH, benign prostate hyperplasia; CIRT, carbon-ion radiotherapy.

**Figure 3 cancers-17-03039-f003:**
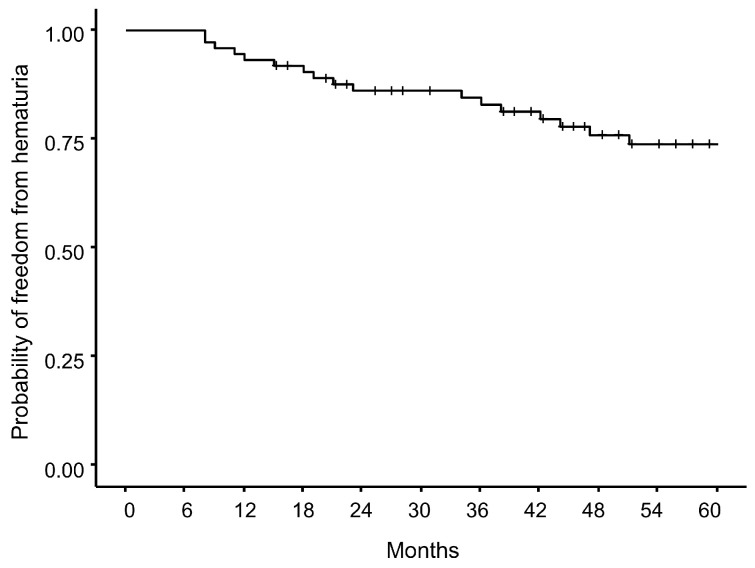
Kaplan–Meier estimates of freedom from hematuria after carbon-ion radiotherapy in patients with prior BPH surgery. The 5-year hematuria-free probability was approximately 75%, with a gradual decline observed over time.

**Figure 4 cancers-17-03039-f004:**
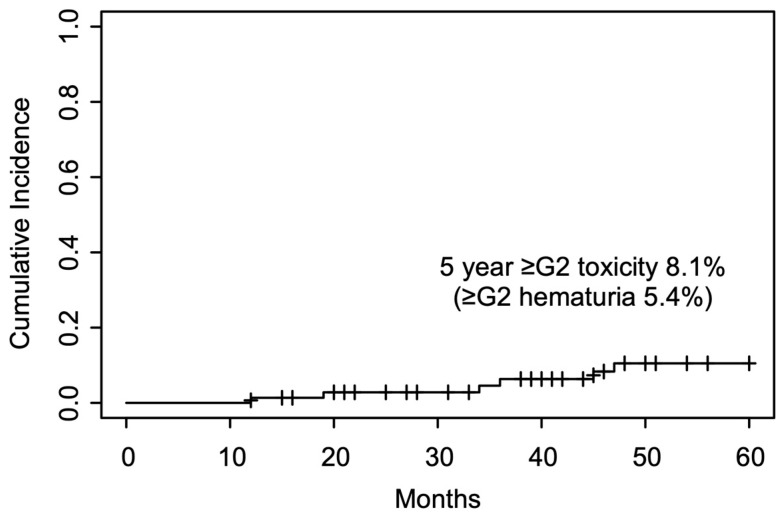
Cumulative incidence of Grade ≥ 2 GU AEs after CIRT in patients with prior BPH surgery. Events included hematuria, dysuria, urinary frequency, and urinary retention. Rates at 12, 24, and 36 months were 0.0%, 1.4%, and 5.0%, respectively. GU, genitourinary; AEs, adverse events; BPH, benign prostatic hyperplasia; CIRT, carbon-ion radiotherapy.

**Figure 5 cancers-17-03039-f005:**
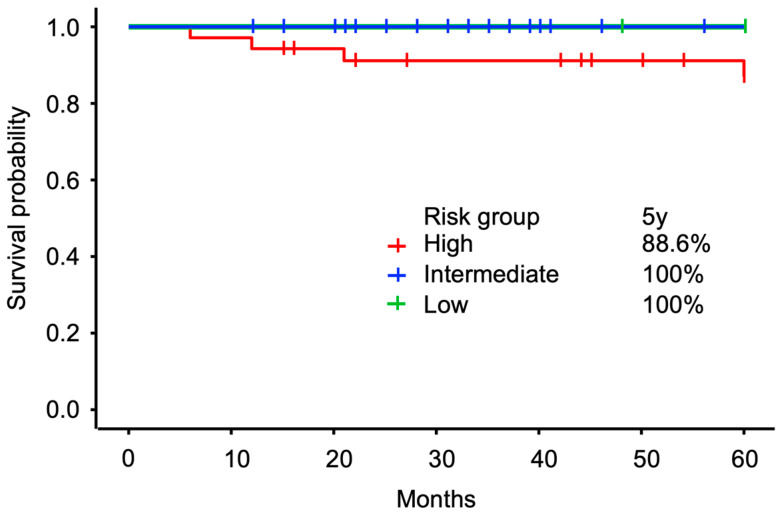
Kaplan–Meier curves of bRFS post-CIRT stratified by NCCN risk group. The 5-year bRFS was 100% in the low- and intermediate-risk groups, and 88.6% [0.749 and 0.991], in the high-risk group. bRFS, biochemical recurrence-free survival; CIRT, carbon-ion radiation therapy; NCCN, National Comprehensive Cancer Network.

**Table 1 cancers-17-03039-t001:** Patient characteristics.

Factors	n (%), Median [IQR]
Follow-up period (months)	60 [48.0–70.75]
Age at CIRT (years)	75.5 [73.0–79.0]
Age at BPH surgery (years)	67 [63.0–72.0]
Interval from BPH surgery to CIRT	8 [3.0–14.0]
Previous BPH surgical treatment	
TURP	49 (66.2%)
HoLEP	15 (20.3%)
TUEB	3 (4.1%)
PVP	3 (4.1%)
SPP	3 (4.1%)
TUMT	1 (1.4%)
(SPP + TURP)	1 (1.4%)
(TURP + HoLEP)	2 (2.7%)
T stage	
T1c	5 (6.8%)
T2a	31 (41.9%)
T2b	10 (14.0%)
T2c	10 (14.0%)
T3a	13 (17.6%)
T3b	3 (4.1%)
Tx	2 (2.7%)
Pretreatment PSA (ng/mL)	7.7 [5.7–13.1]
<10	48 (64.9%)
≥10 and ≤20	14 (18.9%)
>20	12 (16.2%)
Gleason Score	
6	6 (8.1%)
7	43 (58.1%)
8	10 (13.5%)
9	15 (20.3%)
Risk group (NCCN)	
Low	3 (4.1%)
Intermediate	36 (48.6%)
High	35 (47.3%)
Dose fractionation	
57.6 Gy in 16 fractions	15 (20.3%)
51.6 Gy in 12 fractions	49 (66.2%)
54.0 Gy in 12 fractions	10 (13.5%)
Comorbidities	
Diabetes mellitus	9 (12.2%)
Anticoagulant therapy	13 (17.6%)

CIRT, carbon-ion radiotherapy; BPH, benign prostatic hyperplasia; IQR, interquartile range; TURP, transurethral resection of the prostate; HoLEP, holmium laser enucleation of the prostate; TUEB, transurethral enucleation using bipolar energy; PVP, photoselective vaporization of the prostate; SPP, simple suprapubic prostatectomy; NCCN, National Comprehensive Cancer Network; TUMT, transurethral microwave therapy; n, number.

**Table 2 cancers-17-03039-t002:** Acute and late AEs.

	Grade 0–1	Grade 2	Grade 3
Acute AEs			
GU	94.6%	5.4%	0%
GI	100%	0%	0%
Late AEs			
GU	91.8%	6.8%	1.4%
GI	98.6%	1.4%	0%

GU, genitourinary; GI, gastrointestinal; AEs, adverse events. AEs were graded according to CTCAE v5.0.

**Table 3 cancers-17-03039-t003:** Detailed acute and late AEs.

	Grade 2	Grade 3
Acute GU AEs		
Urinary frequency	4.1%	0%
Urinary tract pain	1.4%	0%
Late GU AEs		
Hematuria	4.1%	1.4%
Urinary frequency	1.4%	0%
Urethral stricture	1.4%	0%
Late GI AEs		
Rectal hemorrhage	1.4%	0%

GU: genitourinary; GI, gastrointestinal; AE, adverse events.

**Table 4 cancers-17-03039-t004:** Logistic regression analyses for predictors of Grade ≥ 1 and ≥2 hematuria after CIRT.

	G1 ≥ Hematuria				G2 ≥ Hematuria			
	Univariate Analysis		Multivariate Analysis		Univariate Analysis		Multivariate Analysis	
Variable	OR (95% CI)	*p*-Value	OR (95% CI)	*p*-Value	OR (95% CI)	*p*-Value	OR (95% CI)	*p*-Value
Age at CIRT (years)	0.99 (0.89–1.1)	0.887	1.10 (0.96–1.12)	0.185	1.01 (0.83–1.23)	0.888	1.03 (0.81–1.30)	0.804
Interval from BPH surgery to CIRT	0.93 (0.85–1.02)	0.126	0.87 (0.75–0.98)	* 0.041	0.99 (0.85–0.92)	0.918	0.95 (0.75–1.16)	0.651
Diabetes mellitus	1.67 (0.37–7.48)	0.505	3.47 (0.75–27.2)	0.219	9.0 (1.09–74.2)	* 0.041	9.64 (1.46–14.7)	0.068
Anticoagulant therapy	1.49 (0.40–5.59)	0.553	2.26 (0.44–11.6)	0.318	5.36 (0.68–42.2)	0.111	4.63 (3.12–90.1)	0.252
Risk group: intermediate	1.76 (0.60–5.45)	0.310	2.57 (0.66–11.3)	0.187	0.97 (0.01–8.48)	0.977	1.51 (1.20–2.50)	0.749
Risk group: low	0.00 (NA)	0.991	0.00 (NA)	0.994	0.00 (NA)	0.995	0.00 (NA)	0.997
Surgery: TURP	2.6 (0.61–18.0)	0.246	10.3 (1.41–127)	* 0.037	0.913 (0.11–19.3)	0.939	2.95 (9.30–242)	0.571
Total dose	1.15 (0.94–1.42)	0.184	1.20 (0.92–1.57)	0.185	0.992 (0.65–1.52)	0.972	1.03 (0.56–1.65)	0.917

OR, odds ratio; CI, confidence interval; CIRT, carbon-ion radiotherapy; BPH, benign prostatic hyperplasia; TURP, transurethral resection of the prostate; NA, not applicable. * *p* < 0.05 (statistically significant).

## Data Availability

The data underlying this article will be shared upon reasonable request with the corresponding authors.

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
