# Peer review of "Carbon-Ion Radiotherapy for Prostate Cancer in Patients with a History of Surgery for Benign Prostatic Hyperplasia"

_cancers, 2025, doi:10.3390/cancers17183039_

Round 1
Reviewer 1 Report
Comments and Suggestions for Authors
First observation:
Line 166
Describe AEs – it is the first time the term appears in the text – then put the abbreviation in brackets. Then, you can use the abbreviation. You describe the abbreviation in the table 2 Legend
Second observation:
The topic you chose to analyze is very interesting.
The second observation:
You, correctly and thoroughly, assessed a complex set of parameters related to the use of Carbon-ion radiotherapy (CIRT) in patients with prostate cancer that had a history of surgery for benign prostatic hyperplasia (BPH).
And, then, you stopped!
Further you discussed about these parameters in the context of other researches on the topic.
Finally, you drew the conclusion that the therapeutical technique is good.
In my opinion, the study is incomplete and you cannot draw your final conclusion from this unfinished study.
My suggestion is that your study should contain in the first place the analysis of a control group including patients with prostate cancer but without a history of surgery for BPH treated with CIRT.
That means that you had to compare the efficacy of CIRT related to the presence or absence of a history of surgery for BPH in order to come to a real conclusion.
Then, logically, the reporting to the literature should have been done to studies that made the same comparison.
Thus, you can draw a conclusion about the efficacy of CIRT in different groups of patients with prostate cancer that comes from your study.
So, finally, my advice is to reconsider the work and rewrite the article so it makes sense.
Doing this, your article, otherwise with an interesting and challenging topic, will be good for publishing.
Author Response
We sincerely thank you for the careful evaluation of our manuscript and for providing insightful and constructive comments. These suggestions have been invaluable in improving the clarity, accuracy, and overall quality of our study.
Comment 1 (Line 166):
Describe AEs – it is the first time the term appears in the text – then put the abbreviation in brackets. Then, you can use the abbreviation. You describe the abbreviation in the Table 2 legend.
Response 1:
Thank you for this helpful comment. We considered your suggestion and decided that introducing the full term “adverse events (AEs)” in the section title at its first appearance (page 5, line 180) would provide the clearest presentation for readers. Therefore, we revised the title to explicitly state “Adverse Events (AEs)”, after which we consistently use the abbreviation throughout the manuscript.
Updated text (page 5, line 180):
Comment 2:
The topic you chose to analyze is very interesting. You correctly and thoroughly assessed a complex set of parameters related to the use of CIRT in patients with prostate cancer and a history of surgery for BPH. However, you stopped. In my opinion, the study is incomplete and you cannot draw your final conclusion from this unfinished study. My suggestion is that your study should contain in the first place the analysis of a control group including patients with prostate cancer but without a history of surgery for BPH treated with CIRT. That means that you had to compare the efficacy of CIRT related to the presence or absence of a history of surgery for BPH in order to come to a real conclusion. Then, logically, the reporting to the literature should have been done to studies that made the same comparison. Thus, you can draw a conclusion about the efficacy of CIRT in different groups of patients with prostate cancer that comes from your study. So, finally, my advice is to reconsider the work and rewrite the article so it makes sense. Doing this, your article, otherwise with an interesting and challenging topic, will be good for publishing.
Response 2:
We sincerely appreciate your thoughtful and constructive feedback. We fully agree that including a control group of patients without prior BPH surgery would strengthen the study and allow direct comparison of efficacy. Unfortunately, such a comparison was not feasible in this retrospective single-institution cohort because of differences in baseline characteristics and the challenge of balancing confounders. Nevertheless, we have acknowledged this important limitation more explicitly in the revised Discussion and Conclusions.
Specifically, we have clarified that while our data confirm the safety and feasibility of CIRT in this surgically altered population, the absence of a non-BPH control group limits the generalizability of oncological conclusions. We also emphasized that validation in larger, multicenter prospective studies that incorporate both BPH and non-BPH cohorts will be required to definitively establish the comparative efficacy of CIRT.
Updated text (page 12, Discussion, lines 389–393):
Updated text (page 12, Conclusions, lines 402–404):
Reviewer 2 Report
Comments and Suggestions for Authors
In this manuscript, authors performed aimed to evaluate the long-term safety and oncologic outcomes of Carbon-Ion Radiothterapy for in prostate cancer patients with prior surgery for benign prostatic hyperplasia. Manuscript is interesting however some comments are mandatory.
1)Authors correctly underlined the major limitations of this study (retrospective study, single-institution, short follow up) then conclusions should be considered carefully.
2)Moreover, complications such as haematuria and urethral strictures would be higher rates.
3) Bowel disorders seem so low. Did you consider the spacers? I would read and add a brief comment about them.
Read and add these two references:
- Folkert MR et al. Bowel Disorder Incidence and Rectal Spacer Use in Patients With Prostate Cancer Undergoing Radiotherapy. JAMA Netw Open. 2025 Mar 3;8(3):e250491. doi: 10.1001/jamanetworkopen.2025.0491. PMID: 40067300
- Wong CH et al. Does biodegradable peri-rectal spacer mitigate treatment toxicities in radiation therapy for localised prostate cancer-a systematic review and meta-analysis. Prostate Cancer Prostatic Dis. 2025 Mar 27. doi: 10.1038/s41391-025-00961-0. PMID: 40148672
4) And what about the PVR?
5) Did you evaluate the decision regret about RT? It is an actual theme. I would read and add this reference: Guercio A et al. Patient satisfaction and decision regret in patients undergoing radical prostatectomy: a multicenter analysis. Int Urol Nephrol. 2025 Apr 17. doi: 10.1007/s11255-025-04510-5. PMID: 40246765
6) I would read and add two of the most important manuscripts on the topic in the references:
-Slevin F et al. A Systematic Review of the Efficacy and Toxicity of Brachytherapy Boost Combined with External Beam Radiotherapy for Nonmetastatic Prostate Cancer. Eur Urol Oncol. 2024 Aug;7(4):677-696. doi: 10.1016/j.euo.2023.11.018. Epub 2023 Dec26.PMID: 38151440
- Meng Y et al. Evaluation of the safety and efficacy of stereotactic body radiotherapy for radio-recurrent prostate cancer: a systematic review and meta-analysis. Prostate Cancer Prostatic Dis. 2024 Dec 20. doi: 10.1038/s41391-024-00927-8. PMID: 39702471
Author Response
We sincerely thank the reviewer for the thoughtful and constructive comments, which have helped us to improve our manuscript. We have addressed each point in detail below.
Comment 1:
Authors correctly underlined the major limitations of this study (retrospective study, single-institution, short follow-up); then conclusions should be considered carefully.
Response 1:
We agree with this important comment. In the revised Discussion and Conclusions, we have emphasized more clearly that the retrospective single-institution design, small sample size, relatively short follow-up, and absence of a non-BPH control group necessitate cautious interpretation of our findings. We have also stressed the need for validation in larger prospective multicenter cohorts.
Updated text (page 12, Discussion, lines 389–393):
Updated text (page 12, Conclusions, lines 402–404):
Comment 2:
Moreover, complications such as haematuria and urethral strictures would be higher rates.
Response 2:
Thank you for this observation. In the revised Results and Discussion, we clarified that the incidence of hematuria and urethral stricture was indeed modestly higher than that reported in large-scale CIRT registries, and we discussed potential explanations, including the impact of surgical history, tissue damage patterns after TURP, and patient age.
Updated text (page 10, Discussion, lines 318–319):
Comment 3:
Bowel disorders seem so low. Did you consider the spacers? I would read and add a brief comment about them. References suggested:
- Folkert MR et al., JAMA Netw Open, 2025.
- Wong CH et al., Prostate Cancer Prostatic Dis., 2025.
Response 3:
We thank the reviewer for raising this important point. In our cohort, rectal toxicity was minimal, with only one case of Grade 2 rectal hemorrhage and no severe events. Although hydrogel spacers were not routinely used during the study period, we agree that they represent an important strategy to mitigate rectal morbidity. Accordingly, we have added a paragraph in the Discussion acknowledging the role of peri-rectal spacers in reducing bowel toxicity, with reference to the suggested recent publications (Folkert et al., Wong et al.).
Updated text (page 11, Discussion, lines 347–352):
Comment 4:
And what about the PVR?
Response 4:
We agree that post-void residual urine volume (PVR) is a clinically relevant parameter. Unfortunately, PVR was not systematically recorded in this retrospective analysis and therefore could not be evaluated. This limitation has now been explicitly mentioned in the Discussion. We have also cited prior EBRT and CIRT studies that reported longitudinal changes in bladder function and prostate volume, underscoring the need for incorporating PVR assessments in future prospective analyses.
Updated text (page 11, Discussion, lines 360–365):
Comment 5:
Did you evaluate the decision regret about RT? It is an actual theme. Suggested reference: Guercio A et al., Int Urol Nephrol, 2025.
Response 5:
Thank you for this excellent suggestion. We did not evaluate decision regret in this retrospective study. However, we fully agree that patient-reported outcomes, including satisfaction and regret, are increasingly important to capture the broader impact of treatment beyond physician-reported toxicities. We have added a statement in the Discussion acknowledging the absence of decision regret assessment and citing the suggested reference (Guercio et al.).
Updated text (page 11, Discussion, lines 357–360):
Comment 6:
I would read and add two of the most important manuscripts on the topic in the references:
- Slevin F et al., Eur Urol Oncol, 2024.
- Meng Y et al., Prostate Cancer Prostatic Dis., 2024.
Response 6:
We appreciate this valuable recommendation. Both suggested systematic reviews have now been incorporated into the Discussion to provide a broader international context on alternative radiotherapy modalities, specifically brachytherapy boost and stereotactic body radiotherapy for recurrent disease. These references further contextualize our findings and underscore the relevance of highly conformal radiotherapy approaches in complex clinical scenarios.
Updated text (page 12, Discussion, lines 383–386):
Updated text (page 12, Discussion, lines 386–388):
Reviewer 3 Report
Comments and Suggestions for Authors
An excellent study for the association of CIRS and previous bph surgery in patients with prostate cancer. The article is well presented and evidence- based.
some minor points for the authors:
- This is a retrospective study, but the authors included studies until 2023. Since this is a study submitted in 2025, an inclusion of patients from 2024-2025 could improve the quality of the results.
- In the methods section, the authors explain the different bph procedures. It would be ideal if they explained the criteria that patients undergo radiation and not surgical treatment eg radical prostatectomy. Is this according to patients preference or following local guidelines, or according to commorbidities etc.
- in the results there would be interesting if complications and results from different bph procedures were presented, apart from turp and holep. Eg was there any worse complications for open simple prostatectomy patients?
Author Response
We sincerely thank the reviewer for the positive feedback and thoughtful minor suggestions, which have helped us further refine our manuscript. Our detailed responses are as follows:
Comment 1:
This is a retrospective study, but the authors included patients until 2023. Since this study is submitted in 2025, an inclusion of patients from 2024–2025 could improve the quality of the results.
Response 1:
We appreciate this insightful suggestion. Our dataset was locked at the end of 2023 to allow sufficient follow-up time and ensure consistency of data collection for analysis. While we recognize the value of including more recent patients, this would have resulted in shorter and heterogeneous follow-up intervals, potentially biasing the safety outcomes. We have clarified this point in the revised Methods and Discussion, and we note that future registry-based analyses (such as the J-CROS multicenter registry) will provide updated results including more recent cases.
Comment 2:
In the Methods section, the authors explain the different BPH procedures. It would be ideal if they explained the criteria that patients undergo radiation and not surgical treatment (e.g., radical prostatectomy). Is this according to patients’ preference, local guidelines, or comorbidities?
Response 2:
Thank you for raising this important point. We have revised the Methods section to clarify the selection criteria for CIRT over radical prostatectomy or other local therapies. At our institution, patients were referred for CIRT when they were considered unsuitable for radical prostatectomy or photon-based radiotherapy at other hospitals, or when they preferred a non-surgical modality. The main reasons included advanced age, significant comorbidities, or unfavorable anatomical conditions after BPH surgery. We have now explicitly described these factors in the Patients and Study Design subsection.
Updated text (page 3, Conclusions, lines 116–120):
Comment 3:
In the Results, it would be interesting if complications and results from different BPH procedures were presented, apart from TURP and HoLEP. For example, was there any worse complication for open simple prostatectomy patients?
Response 3:
We agree with this valuable suggestion. In the revised Results, we have added a brief description of outcomes from patients who underwent less standard procedures, such as simple suprapubic prostatectomy (SPP), photoselective vaporization of the prostate (PVP), and transurethral enucleation using bipolar energy (TUEB). While the number of such patients was small, no severe complications were observed. For example, one patient with prior SPP experienced Grade 1 urinary frequency, and one patient with prior PVP and TUEB developed Grade 1 hematuria. We have added these details to provide a more comprehensive overview.
Updated text (page 6, Conclusions, lines 201–204):
Round 2
Reviewer 1 Report
Comments and Suggestions for Authors
I examined the revised version of your manuscript and your answers to my first suggestions.
The comments you added to the Discussion chapter give now a sense to your paper and make it publishable.
However, consider this as a first step in your research and think about extending the study in the very first future.